# Distributional Policy Optimization:
# An Alternative Approach for Continuous Control

Chen Tessler*, Guy Tennenholtz* and Shie Mannor

chen.tessler@campus.technion.ac.il, guytenn@gmail.com, shie@ee.technion.ac.il
Technion Institute of Technology, Haifa, Israel

## Abstract

We identify a fundamental problem in policy gradient-based methods in continuous control. As policy gradient methods require the agent's underlying probability distribution, they limit policy representation to parametric distribution classes. We show that optimizing over such sets results in local movement in the action space and thus convergence to sub-optimal solutions. We suggest a novel distributional framework, able to represent arbitrary distribution functions over the continuous action space. Using this framework, we construct a generative scheme, trained using an off-policy actor-critic paradigm, which we call the Generative Actor Critic (GAC). Compared to policy gradient methods, GAC does not require knowledge of the underlying probability distribution, thereby overcoming these limitations. Empirical evaluation shows that our approach is comparable and often surpasses current state-of-the-art baselines in continuous domains.

## 1   Introduction

Model-free Reinforcement Learning (RL) is a learning paradigm which aims to maximize a cumulative reward signal based on experience gathered through interaction with an environment [Sutton and Barto, 1998]. It is divided into two primary categories. Value-based approaches involve learning the value of each action and acting greedily with respect to it (i.e., selecting the action with highest value). On the other hand, policy-based approaches (the focus of this work) learn the policy directly, thereby explicitly learning a mapping from state to action.

Policy gradients (PGs) [Sutton et al., 2000b] have been the go-to approach for learning policies in empirical applications. The combination of the policy gradient with recent advances in deep learning has enabled the application of RL in complex and challenging environments. Such domains include continuous control problems, in which an agent controls complex robotic machines both in simulation [Schulman et al., 2015, Haarnoja et al., 2017, Peng et al., 2018] as well as real life [Levine et al., 2016, Andrychowicz et al., 2018, Riedmiller et al., 2018]. Nevertheless, there exists a fundamental problem when PG methods are applied to continuous control regimes. As the gradients require knowledge of the probability of the performed action $P(\mathbf{a} \,|\, \mathbf{s})$, the PG is empirically limited to parametric distribution functions. Common parametric distributions used in the literature include the Gaussian [Schulman et al., 2015, 2017], Beta [Chou et al., 2017] and Delta [Silver et al., 2014, Lillicrap et al., 2015, Fujimoto et al., 2018] distribution functions.

In this work, we show that while the PG is properly defined over parametric distribution functions, it is prone to converge to sub-optimal exterma (Section 3). The leading reason is that these distributions are not convex in the distribution space[1] and are thus limited to local improvement in the

action space itself. Inspired by Approximate Policy Iteration schemes, for which convergence guarantees exist [Puterman and Brumelle, 1979], we introduce the Distributional Policy Optimization (DPO) framework in which an agent's policy evolves towards a *distribution* over improving actions. This framework requires the ability to minimize a distance (loss function) which is defined over two distributions, as opposed to the policy gradient approach which requires an explicit differentiation through the density function.

DPO establishes the building blocks for our generative algorithm, the Generative Actor Critic[2]. It is composed of three elements: a generative model which represents the policy, a value, and a critic. The value and the critic are combined to obtain the advantage of each action. A target distribution is then defined as one which improves the value (i.e., all actions with negative advantage receive zero probability mass). The generative model is optimized directly from samples without the explicit definition of the underlying probability distribution using quantile regression and Autoregressive Implicit Quantile Networks (see Section 4). Generative Actor Critic is evaluated on tasks in the MuJoCo control suite (Section 5), showing promising results on several difficult baselines.

## 2 Preliminaries

We consider an infinite-horizon discounted Markov Decision Process (MDP) with a continuous action space. An MDP is defined as the 5-tuple $(\mathcal{S}, \mathcal{A}, P, r, \gamma)$ [Puterman, 1994], where $\mathcal{S}$ is a countable state space, $\mathcal{A}$ the continuous action space, $P : S \times S \times \mathcal{A} \mapsto [0,1]$ is a transition kernel, $r : S \times A \to [0,1]$ is a reward function, and $\gamma \in (0,1)$ is the discount factor. Let $\pi : \mathcal{S} \mapsto \mathcal{B}(\mathcal{A})$ be a stationary policy, where $\mathcal{B}(\mathcal{A})$ is the set of probability measures on the Borel sets of $\mathcal{A}$. We denote by $\Pi$ the set of stationary stochastic policies. In addition to $\Pi$, often one is interested in optimizing over a set of parametric distributions. We denote the set of possible distribution parameters by $\Theta$ (e.g., the mean $\mu$ and variance $\sigma$ of a Gaussian distribution).

Two measures of interest in RL are the value and action-value functions $v^\pi \in \mathbb{R}^{|\mathcal{S}|}$ and $Q^\pi \in \mathbb{R}^{|\mathcal{S}| \times |\mathcal{A}|}$, respectively. The value of a policy $\pi$, starting at state $\mathbf{s}$ and performing action $\mathbf{a}$ is defined by $Q^\pi(\mathbf{s}, \mathbf{a}) = \mathbb{E}^\pi \left[ \sum_{t=0}^{\infty} \gamma^t r(\mathbf{s}_t, \mathbf{a}_t) \mid \mathbf{s}_0 = \mathbf{s}, \mathbf{a}_0 = \mathbf{a} \right]$. The value function is then defined by $v^\pi = \mathbb{E}^\pi [Q^\pi(\mathbf{s}, \mathbf{a})]$. Given the action-value and value functions, the advantage of an action $\mathbf{a} \in \mathcal{A}$ at state $\mathbf{s} \in \mathcal{S}$ is defined by $A^\pi(\mathbf{s}, \mathbf{a}) = Q^\pi(\mathbf{s}, \mathbf{a}) - v^\pi(\mathbf{s})$. The optimal policy is defined by $\pi^* = \arg\max_{\pi \in \Pi} v^\pi$ and the optimal value by $v^* = v^{\pi^*}$.

## 3 From Policy Gradient to Distributional Policy Optimization

Current practical approaches leverage the Policy Gradient Theorem [Sutton et al., 2000b] in order to optimize a policy, which updates the policy parameters according to

$$\theta_{k+1} = \theta_k + \alpha_k \mathbb{E}_{\mathbf{s} \sim d(\pi_{\theta_k})} \mathbb{E}_{\mathbf{a} \sim \pi_{\theta_k}(\cdot \mid \mathbf{s})} \nabla_\theta \log \pi_\theta(\mathbf{a} \mid \mathbf{s}) \mid_{\theta=\theta_k} Q^{\pi_{\theta_k}}(\mathbf{s}, \mathbf{a}), \qquad (1)$$

where $d(\pi)$ is the stationary distribution of states under $\pi$. Since this update rule requires knowledge of the log probability of each action under the current policy $\log \pi_\theta(\mathbf{a} \mid \mathbf{s})$, empirical methods in continuous control resort to parametric distribution functions. Most commonly used are the Gaussian [Schulman et al., 2017], Beta [Chou et al., 2017] and deterministic Delta [Lillicrap et al., 2015] distribution functions. However, as we show in Proposition 1, this approach is not ensured to converge, even though there exists an optimal policy which is deterministic (i.e., Delta) - a policy which is contained within this set.

The sub-optimality of uni-modal policies such as Gaussian or Delta distributions does not occur due to the limitation induced by their parametrization (e.g., the neural network), but is rather a result of the predefined set of policies. As an example, consider the set of Delta distributions. As illustrated in Figure 1, while this set is convex in the parameter $\mu$ (the mean of the distribution), it is not convex in the set $\Pi$. This is due to the fact that $(1-\alpha)\delta_{\mu_1} + \alpha\delta_{\mu_2}$ results in a stochastic distribution over two supports, which cannot be represented using a single Delta function. Parametric distributions such as Gaussian and Delta functions highlight this issue, as the policy gradient considers the gradient w.r.t. the parameters $\mu, \sigma$. This results in local movement in the action space. Clearly such an approach can only guarantee convergence to a locally optimal solution and not a global one.

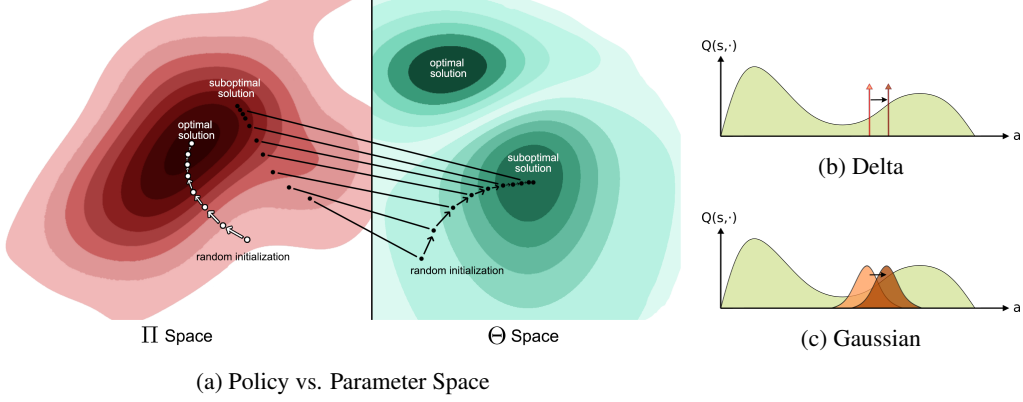

(a) Policy vs. Parameter Space

(b) Delta

(c) Gaussian

Figure 1: (a): A conceptual diagram comparing policy optimization in parameter space $\Theta$ (black dots) in contrast to distribution space $\Pi$ (white dots). Plots depict $Q$ values in both spaces. As parameterized policies are non-convex in the distribution space, they are prone to converge to a local optima. Considering the entire policy space ensures convergence to the global optima. (b,c): Policy evolution of Delta and Gaussian parameterized policies for multi-modal problems.

**Proposition 1.** *For any initial Gaussian policy $\pi_0 \sim \mathcal{N}(\mu_0, \Sigma)$ and $L \in [0, \frac{v^*}{2})$ there exists an MDP $\mathcal{M}$ such that $\pi_\infty$ satisfies*

$$\|v^* - v^{\pi_\infty}\|_\infty > L, \tag{2}$$

*where $\pi_\infty$ is the convergent result of a PG method with step size bounded by $\alpha$. Moreover, given $\mathcal{M}$ the result follows even when $\mu_0$ is only known to lie in some ball of radius $R$ around $\tilde{\mu}_0$, $B_R(\tilde{\mu}_0)$.*

*Proof sketch.* For brevity we prove for the case of $\mathbf{a} \in \mathbb{R}$, such that $B_R$ is a finite interval $[a, b]$. We also assume $[a, b] \subseteq [\mu_0 - 2\alpha, \mu_0 + 2\alpha]$, and $\sigma \to 0$. The general case proof can be found in the supplementary material. Let $\epsilon > 0$. We consider a single state MDP (i.e., x-armed bandit) with action space $\mathcal{A} = \mathbb{R}$ and a multi-modal reward function (similar to the illustration in Figure 1b), defined by

$$r(\mathbf{a}) = \left|\cos\left(\frac{2\pi}{8\alpha}(\mathbf{a} - \mu_0)\right)\right| \left(\epsilon W_{\mu_0 - 2\alpha, \mu_0 + 2\alpha} + (1 - \epsilon)W_{\mu_0 + 2\alpha, \mu_0 + 6\alpha}\right),$$

where $W_{x,y}(z) = \begin{cases} 1 & z \in [x, y] \\ 0 & \text{else} \end{cases}$ is the window function.

In PG, we assume $\mu$ is parameterized by some parameters $\theta$. Without loss of generality, let us consider the derivative with respect to $\theta = \mu$. At iteration $k$ the derivative can be written as $\frac{d}{d\mu} \log \pi_\mu(\mathbf{a})|_{\mu=\mu_k} = -\frac{1}{2\sigma^2}(\mu_k - \mathbf{a})$. PG will thus update the policy parameter $\mu$ by $\mu_{k+1} = \mu_k + \alpha_k \left\{\mathbb{E}_{\mathbf{a}\sim\mathcal{N}(\mu_k,\sigma)}\frac{1}{2\sigma^2}(\mathbf{a} - \mu_k)r(\mathbf{a})\right\}$. As $\sigma \to 0$, it holds that $\text{sign}\left\{\mathbb{E}_{\mathbf{a}\sim\mathcal{N}(\mu_k,\sigma)}(\mathbf{a} - \mu_k)r(\mathbf{a})\right\} = \text{sign}\left\{\frac{d}{d\mathbf{a}}r(\mathbf{a})|_{\mathbf{a}=\mu_k}\right\}$. It follows that if $\epsilon < \frac{1}{3}$ and $\mu_k \in [\mu_0 - 2\alpha, \mu_0 + 2\alpha]$ then so is $\mu_{k+1}$. Then, $\mu_\infty \in [\mu_0 - 2\alpha, \mu_0 + 2\alpha]$. That is, the policy can never reach the interval $[\mu_0 + 2\alpha, \mu_0 + 6\alpha]$ in which the optimal solution lies. Hence, $\|v^* - v^{\pi_\infty}\|_\infty = 1 - 2\epsilon$ and the result follows for $\epsilon < \frac{1}{3}$. $\square$

### 3.1 Distributional Policy Optimization (DPO)

In order to overcome issues present in parametric distribution functions, we consider an alternative approach. In our solution, the policy does not evolve based on the gradient w.r.t. distribution parameters (e.g., $\mu, \sigma$), but rather updates the policy distribution according to

$$\pi_{k+1} = \Gamma\left(\pi_k - \alpha_k \nabla_\pi d(\mathcal{D}_{I^{\pi_k}}^{\pi_k}, \pi)|_{\pi=\pi_k}\right),$$

where $\Gamma$ is a projection operator onto the set of distributions, $d : \Pi \times \Pi \to [0, \infty)$ is a distance measure (e.g., Wasserstein distance), and $\mathcal{D}_{I^\pi}^\pi(\mathbf{s})$ is a distribution defined over the support $I^\pi(\mathbf{s}) = \{\mathbf{a} : A^\pi(\mathbf{s}, \mathbf{a}) > 0\}$ (i.e., the positive advantage). Table 1 provides examples of such distributions.

---

**Algorithm 1** Distributional Policy Optimization (DPO)

---

1: Input: learning rates $\alpha_k \gg \beta_k \gg \delta_k$

2: $\pi_{k+1} = \Gamma\left(\pi_k - \alpha_k \nabla_\pi d(\mathcal{D}^{\pi'_k}_{I^{\pi'_k}}, \pi)\mid_{\pi=\pi_k}\right)$

3: $Q^{\pi'}_{k+1}(\mathbf{s}, \mathbf{a}) = Q^{\pi'}_k(\mathbf{s}, \mathbf{a}) + \beta_k\left(r(\mathbf{s}, \mathbf{a}) + \gamma v^{\pi'}_k(\mathbf{s}) - Q^{\pi'}_k(\mathbf{s}, \mathbf{a})\right)$

4: $v^{\pi'}_{k+1}(\mathbf{s}) = v^{\pi'}_k + \beta_k \int_\mathcal{A}\left(Q^{\pi'}_k(\mathbf{s}, \mathbf{a}) - v^{\pi'}_k(\mathbf{s})\right)$

5: $\pi'_{k+1} = \pi'_k + \delta_k(\pi_k - \pi'_k)$

---

Table 1: Examples of target distributions over the set of improving actions

| Argmax | $\mathcal{D}^\pi_{I^\pi(\mathbf{s})}(\mathbf{a}\mid\mathbf{s}) = \delta_{\arg\max_{a\in I(\pi)} A^\pi(\mathbf{s},\mathbf{a})}(\mathbf{a}\mid\mathbf{s})$ |
|---|---|
| Linear | $\mathcal{D}^\pi_{I^\pi(\mathbf{s})}(\mathbf{a}\mid\mathbf{s}) = \mathbf{1}_{\{\mathbf{a}\in I^\pi\}} \frac{A^\pi(\mathbf{s},\mathbf{a})}{\int_{I^\pi(\mathbf{s})} A^\pi(\mathbf{s},\mathbf{a}')d\mathbf{a}'}$ |
| Boltzmann ($\beta > 0$) | $\mathcal{D}^\pi_{I^\pi(\mathbf{s})}(\mathbf{a}\mid\mathbf{s}) = \mathbf{1}_{\{\mathbf{a}\in I^\pi\}} \frac{\exp\left(\frac{1}{\beta}A^\pi(\mathbf{s},\mathbf{a})\right)}{\int_{I^\pi(\mathbf{s})} \exp\left(\frac{1}{\beta}A^\pi(\mathbf{s},\mathbf{a}')\right)d\mathbf{a}'}$ |
| Uniform | $\mathcal{D}^\pi_{I^\pi(\mathbf{s})}(\mathbf{a}\mid\mathbf{s}) = \text{Uniform}(I^\pi(\mathbf{s}))$ |

Algorithm 1 describes the Distributional Policy Optimization (DPO) framework as a three time-scale approach to learning the policy. It can be shown, under standard stochastic approximation assumptions [Borkar, 2009, Konda and Tsitsiklis, 2000, Bhatnagar and Lakshmanan, 2012, Chow et al., 2017], to converge to an optimal solution. DPO consists of 4 elements: (1) A policy $\pi$ on a fast timescale, (2) a delayed policy $\pi'$ on a slow timescale, (3) a value and (4) a critic, which estimate the quality of the delayed policy $\pi'$ on an intermediate timescale. Unlike the PG approach, DPO does not require access to the underlying p.d.f. In addition, $\pi$ which is updated on the fast timescale views the delayed policy $\pi'$, the value and critic as quasi-static, and as such it can be optimized using supervised learning techniques[3]. Finally, we note that in DPO, the target distribution $\mathcal{D}^{\pi'}_{I^{\pi'}}$ induces a higher value than the current policy $\pi'$, ensuring an always improving policy.

The concept of policy evolution using positive advantage is depicted in Figure 2. While the policy starts as a uni-modal distribution, it is not restricted to this subset of policies. As the policy evolves, less actions have positive advantage, and the process converges to an optimal solution. In the next section we construct a practical algorithm under the DPO framework using a generative actor.

## 4 Method

In this section we present our method, the Generative Actor Critic, which learns a policy based on the Distributional Policy Optimization framework (Section 3). Distributional Policy Optimization requires a model which is both capable of representing arbitrarily complex distributions and can be optimized by minimizing a distributional distance. We consider the Autoregressive Implicit Quantile Network [Ostrovski et al., 2018], which is detailed below.

### 4.1 Quantile Regression & Autoregressive Implicit Quantile Networks

As seen in Algorithm 1, DPO requires the ability to minimize a distance between two distributions. The Implicit Quantile Network (IQN) [Dabney et al., 2018a] provides such an approach using the Wasserstein metric. The IQN receives a quantile value $\tau \in [0, 1]$ and is tasked at returning the value of the corresponding quantile from a target distribution. As the IQN learns to predict the value of the quantile, it allows one to sample from the underlying distribution (i.e., by sampling $\tau \sim U([0, 1])$ and performing a forward pass). Learning such a model requires the ability to estimate the quantiles. The quantile regression loss [Koenker and Hallock, 2001] provides this ability. It is given by $\rho_\tau(u) = (\tau - \mathbf{1}\{u \leq 0\})u$, where $\tau \in [0, 1]$ is the quantile and $u$ the error.

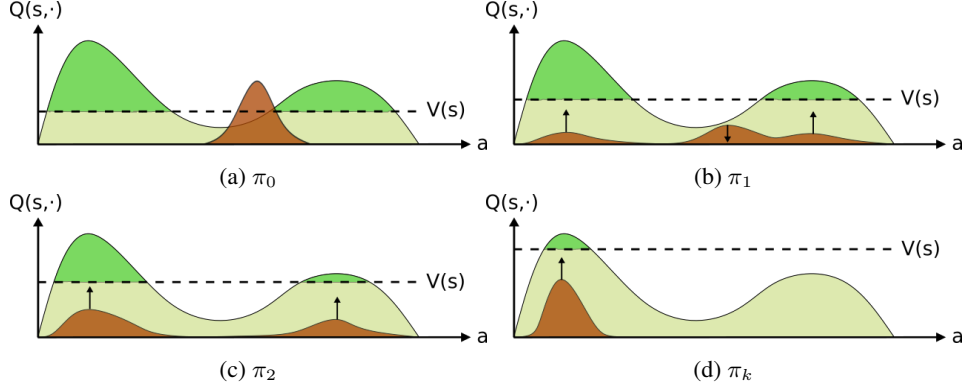

Figure 2: Policy evolution of a general, non-parametric policy, where the target policy is a distribution over the actions with positive advantage. The horizontal dashed line denotes the current value of the policy, the colored green region denotes the target distribution (i.e., the actions with a positive advantage) and $\pi_k$ denotes the policy after multiple updates. As opposed to Delta and Gaussian distributions, the fixed point of this approach is the optimal policy.

Nevertheless, the IQN is only capable of coping with univariate (scalar) distribution functions. Ostrovski et al. [2018] proposed to extend the IQN to the multi-variate case using quantile autoregression [Koenker and Xiao, 2006]. Let $\mathbf{X} = (X_1, \ldots, X_k)$ be an n-dimensional random variable. Given a fixed ordering of the $n$ dimensions, the c.d.f. can be written as the product of conditional likelihoods $F_{\mathbf{X}}(x) = P\left(X^1 \leq x^1, \ldots, X^n \leq x^n\right) = \Pi_{i=1}^n F_{X^i|X^{i-1},\ldots,X^1}(x^i)$. The Autoregressive Implicit Quantile Network (AIQN), receives an i.i.d. vector $\tau \sim U([0,1]^n)$. The network architecture then ensures each output dimension $x_i$ is conditioned on the previously generated values $x_1, \ldots, x_{i-1}$; trained by minimizing the quantile regression loss.

## 4.2 Generative Actor Critic (GAC)

Next, we introduce a practical implementation of the DPO framework. As shown in Section 3, DPO is composed of 4 elements: an actor, a delayed actor, a value, and an action-value estimator. The Generative Actor Critic (GAC) uses a generative actor trained using an AIQN, as described below. Contrary to parametric distribution functions, a generative neural network acts as a universal function approximator, enabling us to represent arbitrarily complex distributions, as corollary of the following lemma.

**Lemma** (Kernels and Randomization [Kallenberg, 2006]). *Let $\pi$ be a probability kernel from a measurable space $S$ to a Borel space $\mathcal{A}$. Then there exists some measurable function $f : S \times [0,1] \to \mathcal{A}$ such that if $\theta$ is $U(0,1)$, then $f(s, \theta)$ has distribution $\pi(\mathbf{a} \,|\, \mathbf{s})$ for every $\mathbf{s} \in S$.*

**Actor:** DPO defines the actor as one which is capable of representing arbitrarily complex policies. To obtain this we construct a generative neural network, an AIQN. The AIQN learns a mapping from a sampled noise vector $\tau \sim U([0,1]^n)$ to a target distribution.

As illustrated in Figure 3, the actor network contains a recurrent cell which enables sequential generation of the action. This generation schematic ensures the autoregressive nature of the model. Each generated action dimension is conditioned only on the current sampled noise scalar $\tau^i$ and the previous action dimensions $\mathbf{a}^{i-1}, \ldots, \mathbf{a}^1$. In order to train the generative actor, the AIQN requires the ability to produce samples from the target distribution $\mathcal{D}^{\pi'}_{I^{\pi'}}$. Although we are unable to sample from this distribution, given an action, we are able to estimate its probability. An unbiased estimator of the loss can be attained by uniformly sampling actions and then multiplying them by their corresponding weight. More specifically, the weighted autoregressive quantile loss is defined by

$$\sum_{\mathbf{a}_j \sim U(\mathcal{A})} \mathcal{D}^{\pi'}_{I^{\pi'}}(\mathbf{a}_j \,|\, \mathbf{s}) \sum_{i=1}^n \rho^k_{\tau^i_j}(\mathbf{a}^i_j - \pi_\phi(\tau^i_j \,|\, \mathbf{a}^{i-1}_j, \ldots, \mathbf{a}^1_j)), \qquad (3)$$

where $\mathbf{a}^i_j$ is the $i^{th}$ coordinate of action $\mathbf{a}_j$, and $\rho^k_{\tau^i_j}$ is the Huber quantile loss [Huber, 1992, Dabney et al., 2018b]. Estimation of $I^{\pi'}$ in the target distribution is obtained using the estimated advantage.

**Delayed Actor:** The delayed actor, also known as Polyak averaging [Polyak, 1990], is an appealing requirement as it is common in off-policy actor-critic schemes [Lillicrap et al., 2015]. The delayed actor is an additional AIQN $\pi_{\theta'}$, which tracks $\pi_\theta$. It is updated based on $\theta'_{k+1} = (1-\alpha)\theta'_k + \alpha\theta_k$ and is used for training the value and critic networks.

**Value and Action-Value:** While it is possible to train a critic and use its empirical mean w.r.t. the policy as a value estimate, we found it to be noisy, resulting in bad convergence. We therefore train a value network to estimate the expectation of the critic w.r.t. the delayed policy. In addition, as suggested in Fujimoto et al. [2018], we train two critic networks in parallel. During both policy and value updates, we refer to the minimal value of the two critics. We observed that this indeed reduced variance and improved overall performance.

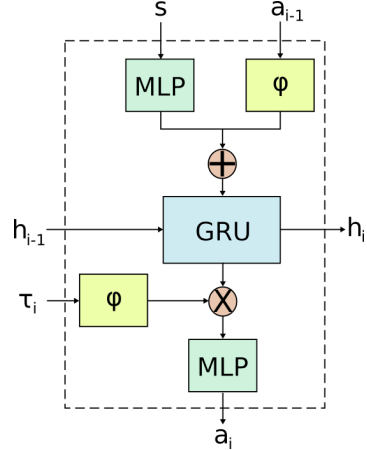

Figure 3: Illustration of the actor's architecture. $\otimes$ is the hadamard product, $\oplus$ a concatenation operator, and $\psi$ a mapping $[0,1] \mapsto \mathbb{R}^d$.

To summarize, GAC combines 4 elements. The delayed actor tracks the actor using a Polyak averaging scheme. The value and critic networks estimate the performance of the delayed actor. Provided $Q$ and $v$ estimations, we are able to estimate the advantage of each action and thus propose the weighted autoregressive quantile loss, used to train the actor network. We refer the reader to the supplementary material for an exhaustive overview of the algorithm and architectural details.

# 5 Experiments

In order to evaluate our approach, we test GAC on a variety of continuous control tasks in the MuJoCo control suite [Todorov et al., 2012]. The agents are composed of $n$ joints: from 2 joints in the simplistic Swimmer task and up to 17 in the Humanoid robot task. The state is a vector representation of the agent, containing the spatial location and angular velocity of each element. The action is a continuous $n$ dimensional vector, representing how much torque to apply to each joint. The task in these domains is to move forward as much as possible within a given time-limit.

We run each task for 1 million steps and, as GAC is an off-poicy approach, evaluate the policy every 5000 steps and report the average over 10 evaluations. We train GAC using a batch size of 128 and uncorrelated Gaussian noise for exploration. Results are depicted in Figure 4. Each curve presented is a product of 5 training procedures with a randomly sampled seed. In addition to our raw results, we compare to the relevant baselines[4], including: (1) DDPG [Lillicrap et al., 2015], (2) TD3 [Fujimoto et al., 2018], an off-policy actor critic approach which represents the policy using a deterministic delta distribution, and (3) PPO [Schulman et al., 2017], an on-policy method which represents the policy using a Gaussian distribution.

As we have shown in the previous sections, DPO and GAC only require *some* target distribution to be defined, namely, a distribution over actions with positive advantage. In our results we present two such distributions: the linear and Boltzmann distributions (see Table 1). We also test a non-autoregressive version of our model [5] using an IQN. For completeness, we provide additional discussion regarding the various parameters and how they performed, in addition to a pseudo-code illustration of our approach, in the supplementary material.

**Comparison to the policy gradient baselines:** Results in Figure 4 show the ability of GAC to solve complex, high dimensional problems. GAC attains competitive results across all domains, often outperforming the baseline policy gradient algorithms and exhibiting lower variance. This is somewhat surprising, as GAC is a vanila algorithm, it is not supported by numerous improvements apparent in recent PG methods. In addition to these results, we provide numerical results in the supplementary material, which emphasize this claim.

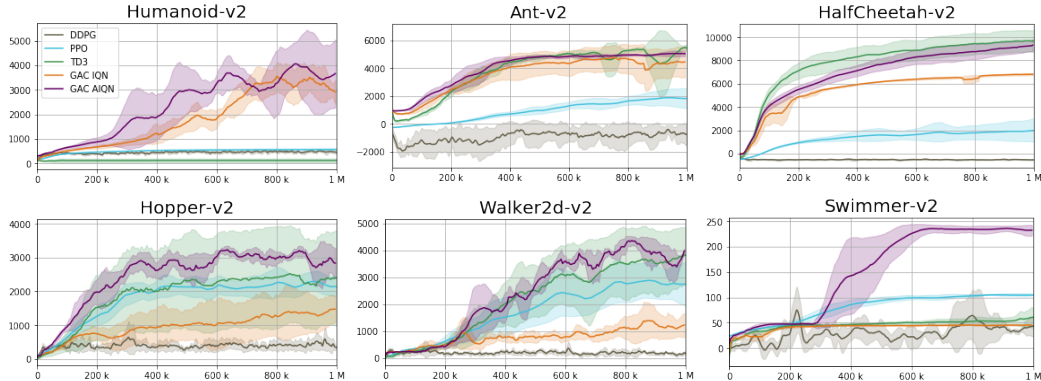

Figure 4: Training curves on continuous control benchmarks. For the Generative Actor Critic approach we present both the Autoregressive and Non-autoregressive approaches, the exact hyperparameters for each domain are provided in the appendix.

Table 2: Relative best GAC results compared to the best policy gradient baseline

| Environment | Humanoid-v2 | Walker2d-v2 | Hopper-v2 | HalfCheetah-v2 | Ant-v2 | Swimmer-v2 |
|---|---|---|---|---|---|---|
| **Relative Result** | **+3447** $(+595\%)$ | **+533** $(+14\%)$ | **+467** $(+17\%)$ | **−381** $(−4\%)$ | **−444** $(−8\%)$ | **+107** $(+81\%)$ |

**Parameter Comparison:** Below we discuss how various parameters affect the behavior of GAC in terms of convergence rates and overall performance:

1. At each step, the target policy is approximated through samples using the weighted quantile loss (Equation (3)). The results presented in Figure 4 are obtained using 32 (256 for HalfCheetah and Walker) samples at each step. 32 (128) samples are taken uniformly over the action space and 32 (128) from the delayed policy $\pi'$ (a form of combining exploration and exploitation). Ablation tests showed that increasing the number of samples improved stability and overall performance. Moreover, we observed that the combination of both sampling methods is crucial for success.

2. Not presented is the Uniform distribution, which did not work well. We believe this is due to the fact that the Uniform target provides an equal weight to actions which are very good while also to those which barely improve the value.

3. We observed that in most tasks, similar to the observations of Korenkevych et al. [2019], the AIQN model outperforms the IQN (non-autoregressive) one.

## 6 Related Work

**Distributional RL:** Recent interest in distributional methods for RL has grown with the introduction of deep RL approaches for learning the distribution of the return. Bellemare et al. [2017] presented the C51-DQN which partitions the possible values $[-v_{\max}, v_{\max}]$ into a fixed number of bins and estimates the p.d.f. of the return over this discrete set. Dabney et al. [2017] extended this work by representing the c.d.f. using a fixed number of quantiles. Finally, Dabney et al. [2018a] extended the QR-DQN to represent the entire distribution using the Implicit Quantile Network (IQN). In addition to the empirical line of work, Qu et al. [2018] and Rowland et al. [2018] have provided fundamental theoretical results for this framework.

**Generative Modeling:** Generative Adversarial Networks (GANs) [Goodfellow et al., 2014] combine two neural networks in a game-theoretic approach which attempt to find a Nash Equilbirium. This equilibrium is found when the generative model is capable of "fooling" the discriminator (i.e., the discriminator is no longer capable of distinguishing between samples produced from the real distribution and those from the generator). Multiple GAN models and training methods have been introduced, including the Wasserstein-GAN [Arjovsky et al., 2017] which minimizes the Wasserstein loss. However, as the optimization scheme is highly non-convex, these approaches are not proven to converge and may thus suffer from instability and mode collapse [Salimans et al., 2016].

**Policy Learning:** Learning a policy is generally performed using one of two methods. The Policy Gradient (PG) [Williams, 1992, Sutton et al., 2000a] defines the gradient as the direction which maximizes the reward under the assumed policy parametrization class. Although there have been a multitude of improvements, including the ability to cope with deterministic policies [Silver et al., 2014, Lillicrap et al., 2015], stabilize learning through trust region updates [Schulman et al., 2015, 2017] and bayesian approaches [Ghavamzadeh et al., 2016], these methods are bounded to parametric distribution sets (as the gradient is w.r.t. the log probability of the action). An alternative line of work formulates the problem as a maximum entropy [Haarnoja et al., 2018], this enables the definition of the target policy using an energy functional. However, training is performed via minimizing the KL-divergence. The need to know the KL-divergence limits practical implementation to parametric distributions functions, similar to PG methods.

## 7  Discussion and Future Work

In this work we presented limitations inherent to empirical Policy Gradient (PG) approaches in continuous control. While current PG methods in continuous control are computationally efficient, they are not ensured to converge to a global extrema. As the policy gradient is defined w.r.t. the log probability of the policy, the gradient results in local changes in the action space (e.g., changing the mean and variance of a Gaussian policy). These limitations do not occur in discrete action spaces.

In order to ensure better asymptotic results, it is often needed to use methods that are more complex and computationally demanding (i.e., "No Free Lunch" [Wolpert et al., 1997]). Existing approaches attempting to mitigate these issues, either enrich the policy space using mixture models, or discretize the action space. However, while the discretization scheme is appealing, there is a clear trade-off between optimality and efficiency. While finer discretization improves guarantees, the complexity (number of discrete actions) grows exponentially in the action dimension [Tang and Agrawal, 2019].

Similar to the limitations inherent in PG approaches, these limitations also exist when considering mixture models, such as Gaussian Mixtures. A mixture model of $k$-Gaussians provides a categorical distribution over $k$ Gaussian distributions. The policy gradient w.r.t. these parameters, similarly to the single Gaussian model, directly controls the mean $\mu$ and variance $\sigma$ of each Gaussian independently. As such, even a mixture model is confined to local improvement in the action space.

In practical scenarios, and as the number of Gaussians grows, it is likely that the modes of the mixture would be located in a vicinity of a global optima. A Gaussian Mixture model may therefore be able to cope with various non-convex continuous control problems. Nevertheless, we note that Gaussian Mixture models, unlike a single Gaussian, are numerically unstable. Due to the summation over Gaussians, the log probability of a mixture of Gaussians does not result in a linear representation. This can cause numerical instability, and thus hinder the learning process. These insights lead us to question the optimality of current PG approaches in continuous control, suggesting that, although these approaches are well understood, there is room for research into alternative policy-based approaches.

In this paper we suggested the Distributional Policy Optimization (DPO) framework and its empirical implementation - the Generative Actor Critic (GAC). We evaluated GAC on a series of continuous control tasks under the MuJoCo control suite. When considering overall performance, we observed that despite the algorithmic maturity of PG methods, GAC attains competitive performance and often outperforms the various baselines. Nevertheless, as noted above, there is "no free lunch". While GAC remains as sample efficient as the current PG methods (in terms of the batch size during training and number of environment interactions), it suffers from high computational complexity.

Finally, the elementary framework presented in this paper can be extended in various future research directions. First, improving the computational efficiency is a top priority for GAC to achieve deployment in real robotic agents. In addition, as the target distribution is defined w.r.t. the advantage function, future work may consider integrating uncertainty estimates in order to improve exploration. Moreover, PG methods have been thoroughly researched and many of their improvements, such as trust region optimization [Schulman et al., 2015], can be adapted to the DPO framework. Finally, DPO and GAC can be readily applied to other well-known frameworks such as the Soft-Actor-Critic [Haarnoja et al., 2018], in which entropy of the policy is encouraged through an augmented reward function. We believe this work is a first step towards a principal alternative for RL in continuous action space domains.

# 8 Acknowledgement

We thank Yonathan Efroni for his fruitful comments that greatly improved this paper.

## Footnotes

[1] As an example, consider the Gaussian distribution, which is known to be non-convex.

[2]Code provided in the following *anonymous* repository: github.com/tesslerc/GAC

[3]Assuming the target distribution is 'fixed', the policy $\pi$ can be trained using a supervised learning loss, e.g., GAN, VAE or AIQN.

[4]We use the implementations of DDPG and PPO from the OpenAI baselines repo [Dhariwal et al., 2017], and TD3 [Fujimoto et al., 2018] from the authors GitHub repository.

[5]Theoretically, the dimensions of the actions may be correlated and thus should be represented using an auto-regressive model.

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
