[Supplementary Material]

# A   Proof of Proposition 1

Let $\epsilon > 0$. We consider a single state MDP (i.e., x-armed bandit) with action space $\mathcal{A} = \mathbb{R}^d$ and a multi-modal reward function defined by

$$r(\mathbf{a}) = \epsilon \delta_{\tilde{\mu}_0}(\mathbf{a}) + (1 - \epsilon)\delta_{\tilde{\mu}_0 + D \cdot \mathbf{1}}(\mathbf{a}),$$

where $D = D(R, \epsilon)$ will be defined later, and $\delta_x(\mathbf{a})$ is the Dirac delta function satisfying $\int_{\mathbf{a}} g(\mathbf{a}) d(\delta_x(\mathbf{a})) = g(x)$ for all continuous compactly supported functions $g$.

Denote by $f_{\mu, \Sigma}(\mathbf{a})$ the multivariate Gaussian distribution, defined by

$$f_{\mu, \Sigma}(\mathbf{a}) = (2\pi |\Sigma|)^{-\frac{k}{2}} e^{-(\mathbf{a} - \mu)^T \Sigma^{-1}(\mathbf{a} - \mu)}.$$

In PG, we assume $\mu$ is parameterized by some parameters $\theta$. Without loss of generality, let us consider the derivative with respect to $\theta = \mu$. At iteration $k$ the derivative can be written as

$$\nabla_\mu \log \pi_\mu(\mathbf{a}) \mid_{\mu = \mu_k} = \Sigma^{-1}(\mathbf{a} - \mu_k).$$

PG will thus update the policy parameter $\mu$ by

$$\mu_{k+1} = \mu_k + \alpha_k \left\{ \mathbb{E}_{\mathbf{a} \sim \mathcal{N}(\mu_k, \Sigma)} \Sigma^{-1}(\mathbf{a} - \mu_k) r(\mathbf{a}) \right\}.$$

Notice that given a Bernoulli random variable $B = \begin{cases} 0 & , \text{w.p. } \epsilon \\ D & , \text{w.p. } 1 - \epsilon \end{cases}$, one can write $r(\mathbf{a}) = \mathbb{E} \delta_{\tilde{\mu}_0 + B \cdot \mathbf{1}}(\mathbf{a})$. Then by Fubini's theorem we have

$$\mathbb{E}_{\mathbf{a} \sim \mathcal{N}(\mu_k, \Sigma)}(\mathbf{a} - \mu_k) r(\mathbf{a})$$
$$= \mathbb{E}_B \mathbb{E}_{\mathbf{a} \sim \mathcal{N}(\mu_k, \Sigma)}(\mathbf{a} - \mu_k) \delta_{\tilde{\mu}_0 + B \cdot \mathbf{1}}(\mathbf{a})$$
$$= \mathbb{E}_B (\tilde{\mu}_0 + B \cdot \mathbf{1} - \mu_k) f_{\mu_k, \Sigma}(\tilde{\mu}_0 + B \cdot \mathbf{1}).$$

We wish to show that the gradient has a higher correlation with the direction of $\tilde{\mu}_0 - \mu_k$ rather than $\tilde{\mu}_0 + D \cdot \mathbf{1} - \mu_k$. That is we wish to show that

$$\left( \mathbb{E}_{\mathbf{a} \sim \mathcal{N}(\mu_k, \Sigma)} \Sigma^{-1}(\mathbf{a} - \mu_k) r(\mathbf{a}) \right)^T \left( \frac{\tilde{\mu}_0 - \mu_k}{\|\tilde{\mu}_0 - \mu_k\|} \right) > \left( \mathbb{E}_{\mathbf{a} \sim \mathcal{N}(\mu_k, \Sigma)} \Sigma^{-1}(\mathbf{a} - \mu_k) r(\mathbf{a}) \right)^T \left( \frac{\tilde{\mu}_0 + D \cdot \mathbf{1} - \mu_k}{\|\tilde{\mu}_0 + D \cdot \mathbf{1} - \mu_k\|} \right).$$

Substituting $r(\mathbf{a})$ the above equation is equivalent to

$$\left( \mathbb{E}_B (\tilde{\mu}_0 + B \cdot \mathbf{1} - \mu_0) f_{\mu_0, \Sigma}(\tilde{\mu}_0 + B \cdot \mathbf{1}) \right)^T \left( \frac{\tilde{\mu}_0 - \mu_k}{\|\tilde{\mu}_0 - \mu_k\|} \right)$$
$$> \left( \mathbb{E}_B (\tilde{\mu}_0 + B \cdot \mathbf{1} - \mu_0) f_{\mu_0, \Sigma}(\tilde{\mu}_0 + B \cdot \mathbf{1}) \right)^T \left( \frac{\tilde{\mu}_0 + D \cdot \mathbf{1} - \mu_k}{\|\tilde{\mu}_0 + D \cdot \mathbf{1} - \mu_k\|} \right). \tag{4}$$

Proving Equation (4) for all $k \geq 0$ will complete the proof.
We continue the proof by induction on $k$.
**Base case (k = 0):**
Recall that $\mu_0 \in B_R(\tilde{\mu}_0)$. Writing Equation (4) explicitly we get

$$\text{LHS} = \epsilon \|\tilde{\mu}_0 - \mu_0\| f_{\mu_0, \Sigma}(\tilde{\mu}_0) + (1 - \epsilon) f_{\mu_0, \Sigma}(\tilde{\mu}_0 + D \cdot \mathbf{1}) (\tilde{\mu}_0 - \mu_0 + D \cdot \mathbf{1})^T \frac{\tilde{\mu}_0 - \mu_0}{\|\tilde{\mu}_0 - \mu_0\|},$$

$$\text{RHS} = \epsilon f_{\mu_0, \Sigma}(\tilde{\mu}_0) (\tilde{\mu}_0 - \mu_0)^T \frac{\tilde{\mu}_0 - \mu_0 + D \cdot \mathbf{1}}{\|\tilde{\mu}_0 - \mu_0 + D \cdot \mathbf{1}\|} + (1 - \epsilon) \|\tilde{\mu}_0 - \mu_0 + D \cdot \mathbf{1}\| f_{\mu_0, \Sigma}(\tilde{\mu}_0 + D \cdot \mathbf{1}).$$

Since $f_{\mu_0, \Sigma}(\tilde{\mu}_0 + D \cdot \mathbf{1}) \propto \exp\{-D \cdot \mathbf{1}\}$ we only need to show that for large enough $D$ (which depends on the constants $\epsilon$ and $R$)

$$\|\tilde{\mu}_0 - \mu_0\| > (\tilde{\mu}_0 - \mu_0)^T \cdot \mathbf{1} \frac{D}{\|\tilde{\mu}_0 - \mu_0 + D \cdot \mathbf{1}\|},$$

as all other values tend to zero.
If $(\tilde{\mu}_0 - \mu_0)^T \mathbf{1} < 0$ then we are done. Otherwise, if $(\tilde{\mu}_0 - \mu_0)^T \mathbf{1} \geq 0$ then

$$(\tilde{\mu}_0 - \mu_0)^T \cdot \mathbf{1} \frac{D}{\|\tilde{\mu}_0 - \mu_0 + D \cdot \mathbf{1}\|} \leq \|\tilde{\mu}_0 - \mu_0\| \frac{D}{\|\tilde{\mu}_0 - \mu_0 + D \cdot \mathbf{1}\|} \leq \|\tilde{\mu}_0 - \mu_0\|,$$

---

**Algorithm 2** Generative Actor Critic

---

1: Input: number of time steps $T$, policy samples $K$, minibatch size $N$
2: Initialize critic networks $Q_{\theta_1}$, $Q_{\theta_2}$, value network $v_\psi$ and actor network $\pi_\phi$ with random parameters $\theta_1, \theta_2, \psi, \phi$
3: Initialize target networks $\theta_1' \leftarrow \theta_1, \theta_2' \leftarrow \theta_2, \psi' \leftarrow \psi, \phi' \leftarrow \phi$
4: Initialize replay buffer $\mathcal{B}$
5: **for** $t = 0, 1, ..., T$ **do**
6:     Select action with exploration noise $\mathbf{a} \sim \pi_\phi(s) + \epsilon$,
7:     $\epsilon \sim \mathcal{N}(0, \sigma)$ and observe reward $r$ and new state $s'$
8:     Store transition tuple $(\mathbf{s}, \mathbf{a}, r, \mathbf{s}')$ in $\mathcal{B}$
9:     Sample mini-batch of $N$ transitions $(\mathbf{s}, \mathbf{a}, r, \mathbf{s}')$ from $\mathcal{B}$
10:     $y_Q \leftarrow r + \gamma v_{\psi'}(\mathbf{s}')$
11:     Update critics:
12:

$$\theta \leftarrow \theta - \frac{1}{N} \nabla_{\theta_i} \sum (y_Q - Q_{\theta_i}(\mathbf{s}, \mathbf{a}))^2$$

13:     $\tilde{\mathbf{a}}_j \leftarrow \pi_{\phi'}(\tau \,|\, \mathbf{s}), \forall 1 \leq j \leq K, \tau \sim U([0,1]^n)$
14:     $y_v \leftarrow \min_{i=1,2} \sum_{j=1}^{K} Q_{\theta_i'}(\mathbf{s}, \tilde{\mathbf{a}}_j)$
15:     Update value:

$$\psi \leftarrow \psi - N^{-1} \nabla_\psi \sum (y_v - v_\psi(\mathbf{s}))^2$$

16:     Sample actions $\hat{\mathbf{a}}_1, \ldots, \hat{\mathbf{a}}_K$ from sampling policy $\sigma(\pi_{\phi'}, \mathcal{A})$
17:     $\hat{\mathcal{A}}_k \leftarrow \{\hat{\mathbf{a}}_j : 1 \leq j \leq K, \min_{i=1,2} Q_{\theta_i'}(\mathbf{s}_k, \hat{\mathbf{a}}_j) > v_{\psi'}(\mathbf{s}_k)\}$
18:     Update actor:

$$\phi \leftarrow \phi - \frac{1}{N} \nabla_\phi \sum_{n=1}^{N} \sum_{\hat{\mathbf{a}} \in \hat{\mathcal{A}}_k} \overset{\text{action dim}}{\sum_{i=1}} \rho_{\tau_i}^k \left( \hat{\mathbf{a}}^i - \pi_\phi(\tau_i | \hat{\mathbf{a}}^{i-1}, \ldots, \hat{\mathbf{a}}^1, \mathbf{s}_k) \right) \mathcal{D}_{I^{\pi_k'}}^{\pi_k'}$$

19:     Update target networks:

$$\theta_i' \leftarrow \tau \theta_i + (1 - \tau)\theta_i'$$
$$\psi' \leftarrow \tau \psi + (1 - \tau)\psi'$$
$$\phi' \leftarrow \tau \phi + (1 - \tau)\phi'$$

---

where in the first step we used the Cauchy–Schwarz inequality, and in the second step we used the fact if a vector $\mathbf{x}$ satisfies $\mathbf{x}^T \mathbf{1} \geq 0$ then for any constant $C > 0$, $\|\mathbf{x} + C \cdot \mathbf{1}\| \geq C$.

**Induction step:**
Assume Equation (4) holds from some $k \geq 0$. Then by the gradient procedure we know that $\mu_k \in B_R(\tilde{\mu}_0)$, and thus we can use the same proof as the base case. Hence, $\|v^* - v^{\pi_\infty}\|_\infty = 1 - 2\epsilon$ and the result follows for $\epsilon < \frac{1}{3}$.

## B   Experimental Details

Our approach is depicted in Algorithm 2. In addition, we provide a numerical comparison of the various approaches in Table 3. These results show a clear picture.

**Target policy estimation:** To estimate the target policy, for each state $\mathbf{s}$, we sample 128 actions uniformly from the action space $\mathcal{A}$, 128 samples from the target policy $\pi_{\phi'}$ and the per-sample loss is weighted by the positive advantage $A(\mathbf{s}, \cdot)^+$. This can be seen as a form of 'exploration-exploitation' - while uniform sampling ensures proper exploration of the action set, sampling from the policy has a higher probability of producing actions with positive advantage.

Table 3: Comparison of the maximal attained value across training.

| Environment | DDPG | TD3 | PPO | GAC AIQN | GAC IQN |
|---|---|---|---|---|---|
| Hopper-v2 | $638 \pm 477$ | $2521 \pm 1429$ | $2767 \pm 421$ | $3234 \pm 122$ | $1473 \pm 421$ |
| Humanoid-v2 | $519 \pm 44$ | $184 \pm 67$ | $579 \pm 30$ | $4056 \pm 878$ | $3547 \pm 572$ |
| Walker2d-v2 | $364 \pm 223$ | $3824 \pm 995$ | $3694 \pm 765$ | $4357 \pm 160$ | $1390 \pm 651$ |
| Swimmer-v2 | $75 \pm 46$ | $60 \pm 20$ | $131 \pm 1$ | $238 \pm 3$ | $45 \pm 0$ |
| Ant-v2 | $-399 \pm 323$ | $5508 \pm 191$ | $2899 \pm 973$ | $5064 \pm 208$ | $4784 \pm 895$ |
| HalfCheetah-v2 | $-395 \pm 81$ | $9681 \pm 908$ | $3787 \pm 2249$ | $9300 \pm 515$ | $6807 \pm 98$ |

Table 4: AIQN Hyperparameters

| | Humanoid-v2, Hopper-v2, Ant-v2, Swimmer-v2 | Walker2d-v2 | HalfCheetah-v2 |
|---|---|---|---|
| Distribution | $\max\{\exp Q(s,a) - v(s), 20\}$ | $\mathrm{softmax}(Q(s,a) - v(s))$ | $Q(s,a) - v(s)$ |
| $\pi$ LR | $1e^{-4}$ | $1e^{-3}$ | $1e^{-3}$ |
| $Q/v$ LR | $1e^{-3}$ | $1e^{-3}$ | $1e^{-3}$ |
| $\pi$ grad clip | 1 | $\infty$ | $\infty$ |
| $Q/v$ grad clip | 5 | $\infty$ | $\infty$ |
| # of samples | 64 | 256 | 256 |

The loss is thus the weighted quantile loss. We do note that while one would want to define the target policy as the linear/Boltzmann distribution over the positive advantage, this is not possible in practice. As actions are sampled, we can only construct such a distribution on a per-batch instance. This approach does provide higher weight for better performing actions, but does result in a different underlying distribution. In addition, in order to ensure stability, we normalize the quantile loss weights in each batch - this is to ensure that very small (high) advantage values do not incur a near-zero (huge) gradients which may harm model stability.

**Architectural Details:**

**Actor:** As presented in Figure 3, our architecture incorporates a recurrent cell. The recurrent cell ensures that each dimension $i$ of the action is a function of the state $\mathbf{s}$, the sampled quantile $\tau_i$ and the previous predicted action dimensions $\mathbf{a}_1, \ldots, \mathbf{a}_{i-1}$. Notice that using this architecture, the prediction of $\mathbf{a}_i$ is not affected by $\tau_1, \ldots, \tau_{i-1}$. This approach is a strict requirement when considering the autoregressive approach.

We believe other, potentially more efficient architectures can be explored. For instance, a fully connected network, similar to the non-autoregressive approach, with attention over the previous action dimensions may work well [Vaswani et al., 2017]. Such evaluation is out of the scope of this work and is an interesting investigation for future work.

**Value & Critic:** While the actor architecture is a non-standard approach, for both the value and critic networks, we use the classic MLP network. Specifically, we use a two layer fully connected network with 400 and 300 neurons in each layer, respectively. Similarly to Fujimoto et al. [2018], the critic receives a concatenated vector of both the state and action as input.

# C  Discussion and Common Mistakes

As shown in the body of the paper, there exist alternative approaches. We take this section in order to provide some additional discussion into how and why we decided on certain approaches and what else can be done.

## C.1 Alternative Gradient Approaches

Going back to the policy gradient approach, specifically the deterministic version, we can write the value of the current policy of our generative model (policy) as:

$$v^\pi(\mathbf{s}) = \int_{\tau \in [0,1]^n} Q(\mathbf{s}, F^{-1}(\mathbf{s}\,|\tau)) d\tau\,,$$

or an estimation using samples

$$v^\pi(\mathbf{s}) = \frac{1}{N} \sum_{i=1}^{N} Q(\mathbf{s}, F^{-1}(\mathbf{s}\,|\tau_i)) \mid_{\tau_i \sim U([0,1]^n)}\,.$$

It may then be desirable to directly optimize this objective function by taking the gradient w.r.t. the parameters of $F^{-1}$. However, this approach **does not ensure optimality**. Clearly, the gradient direction is provided by the critic $Q$ for each value of $\tau$. This can be seen as optimizing an ensemble of DDPG models whereas each $\tau$ value selects a different model from this set. As DDPG is a unimodal parametric distribution and is thus not ensured to converge to an optimal policy, this approach suffers from the same caveats.

However, Evolution Strategies [Salimans et al., 2017] is a feasible approach. As opposed to the gradient method, this approach can be seen as directly calculating $\nabla_\pi v^\pi$, i.e., it estimates the best direction in which to move the policy. As long as the policy is still capable of representing arbitrarily complex distributions this approach should, in theory, converge to a global maxima. However, as there is interest in sample efficient learning, our focus in this work was on introducing an off-policy learning method under the common actor-critic framework.

## C.2 Target Networks and Stability

Our empirical approach, as shown in Algorithm 2, uses a target network for each approximator (critic, value and the target policy). While the critic and value target networks are mainly for stability of the empirical approach, they can be disposed of, the policy target network is required for the algorithm to converge (as shown in Section 3).

The quantile loss, and any distribution loss in general, is concerned with moving probability mass from the current distribution towards the target distribution. This leads to two potential issues when lacking the delayed policy: (1) non-quasi-stationarity of the target distribution, and (2) non-increasing policy.

The first point is important from an optimization point of view. As the quantile loss is aimed to estimate some target distribution, the assumption is that this distribution is static. Lacking the delayed policy network, this distribution potentially changes at each time step and thus can not be properly estimated using sample based approaches. The delayed policy solves this problem, as it tracks the policy on a slower timescale it can be seen as quasi-static and thus the target distribution becomes well defined.

The second point is important from an RL point of view. In general, RL proofs evolve around two concepts - either you are attempting to learn the optimal Q values and convergence is shown through proving the operator is contracting towards a unique globally stable equilibrium, or the goal is to learn a policy and thus the proof is based on showing the policy is monotonically improving. As the delayed policy network slowly tracks the policy network, the multi-timescale framework tells us that "by the time" the delayed policy network changes, the policy network can be assumed to converge. As the policy network is aimed to estimate a distribution over the positive advantage of the delayed policy, this approach ensures that the delayed policy is monotonically improving (under the correct theoretical step-size and realizability assumptions).

## C.3 Sample Complexity and Policy Samples

When considering sample complexity in its simplest form, our approach is as efficient as the baselines we compared to. It does not require the use of larger batches nor does it require more environment samples. However, as we are optimizing a generative model, it does require sampling from the model itself.

As opposed to Dabney et al. [2018a], we found that in our approach the number of samples does affect the convergence ability of the network. While using 16 samples for each transition in the batch did result in relatively good policies, increasing this number affected stability and performance positively. For this reason, we decided to run with a sample size of 128. This results in longer training times. For instance, training the TD3 algorithm on the Hopper-v2 domain using two NVIDIA GTX 1080-TI cards took around 3 hours, whereas our approach took 40 hours to train. We argue that as often the resulting policy is what matters, it is worth to sacrifice time efficiency in order to gain a better final result.

### C.4 Generative Adversarial Policy Training

Our approach used the AIQN framework in order to train a generative policy. An alternative method for learning distributions from samples is using the GAN framework. A discriminator can be trained to differentiate between samples from the current policy and those from the target distribution; thus, training the policy to 'fool' the discriminator will result in generating a distribution similar to the target.

However, while the GAN framework has seen multiple successes, it still lacks the theoretical guarantees of convergence to the Nash equilibrium. As opposed to the AIQN which is trained on a supervision signal, the GAN approach is modeled as a two player zero-sum game.

## D  Distributional Policy Optimization Assumptions

We provide the assumptions required for the 3-timescale stochastic approximation approach, namely DPO, to converge.

The first assumption is regarding the step-sizes. It ensures that the policy moves on the fastest timescale, the value and critic on an intermediate and the delayed policy on the slowest. This enables the quasi-static analysis in which the fast elements see the slower as static and the slow view the faster as if they have already converged.

**Assumption 1.** *[Step size assumption]*

$$\sum_{n=0}^{\infty} \alpha_k = \sum_{n=0}^{\infty} \beta_k = \infty = \sum_{n=0}^{\infty} \delta_k = \infty,$$

$$\sum_{n=0}^{\infty} \left( \alpha_k^2 + \beta_k^2 + \delta_k^2 \right) < \infty,$$

$$\frac{\alpha_k}{\beta_k} \to 0 \ \ and \ \ \frac{\beta_k}{\delta_k} \to 0 \ .$$

The second assumption requires that the action set be compact. Since there exists a deterministic policy which is optimal, this assumption ensures that this policy is indeed finite and thus the process converges.

**Assumption 2.** *[Compact action set] The action set $\mathcal{A}(\mathbf{s})$ is compact for every $\mathbf{s} \in \mathcal{S}$.*

The final two assumptions (3 and 4) ensure that $\pi$, moving on the fast time-scale, converges. The Lipschitz assumption ensures that the action-value function and in turn the target distribution $D_{I^{\pi'}}$ are smooth.

**Assumption 3.** *[Lipschitz and bounded Q] The action-value function $Q^\pi(\mathbf{s}, \cdot)$ is Lipschitz and bounded for every $\pi \in \Pi$ and $\mathbf{s} \in \mathcal{S}$.*

**Assumption 4.** *For any $\mathcal{D} \in \Pi$ and $\theta \in \Theta$, there exists a loss $L$ such that $\nabla_\theta L(\pi_\theta, \mathcal{D}) \to 0$ as $\pi_\theta \to \mathcal{D}$.*

Finally, it can be shown that DPO converges under these assumptions using the standard multi-timescale approach.