[Reviews · NeurIPS 2019]

Reviewer 1



This paper proposes a distributional policy optimization (DPO) framework and its practical implementation, generative actor-critic (GAC) that belongs to off-policy actor-critic methods. Policy gradient methods, which are currently dominant in continuous control problems, are prone to local optima, thus it is valuable to propose a method addressing that problem fundamentally. Overall, the paper is well written and the proposed algorithm seems novel and sound. - In Algorithm 1, it is not clear what (s,a) and s is, for Q(s,a) and V(s). Does it stand for 'every' state-action pair and state, or the state-action pairs that are visited by the current policy \pi_k'? If it corresponds to the latter, it seems that DPO would possibly not converge to the global optima. For example, suppose that the the initial policy is given as the Dirac delta distribution \pi_0(a|s) = \delta_{0}(a), and the (deterministic) transition function is defined as f(s,a) = s + a. Then, with the \pi_0, only the initial state can be visited, thus the value functions and the policy will remain the same and not be updated. Assumptions about behavior policy are not mentioned in the paper. - L109: can can -> can - In L114, what does 'as \pi is updated on the fast timescale, it can be optimized using supervised learning techniques' mean specifically? Please elaborate on the relationship between supervised-learning and fast timescale. - In DPO the delayed policy \pi' is updated by the convex combination of two 'probability distributions', while in GAC the delayed actor is updated by a convex combination of two 'parameters' of each probability distribution. Therefore, DPO and GAC are not perfectly aligned. - For the actor, why did you choose to adopt implicit quantile networks, though GAC does not require 'quantile' estimation? It seems that conditional GAN or conditional VAE could also be possible.

Reviewer 2



The idea proposed in the paper is very interesting. Policy gradient methods are very popular nowadays and this paper propose a method to approach one of their weakness. The paper is clearly written and figures are helpful for the reader. The DPO procedure would have benefited however to be a bit clearer: for example, why is the delayed actor present in this general procedure? It seems to me mainly helpful to help convergence and does not seem an requirement for the method. Besides, although valid theoretically, a three time scale algorithm seems hard to do in practice. #Post-rebuttal update" I appreciate the efforts the authors put into their rebuttal. I will however keep my score as 7, as I vote for accepting this submission, but would not be upset if it was rejected (clarity of the paper could benefit from a restructuration and I am not a huge fan of the 3-time scale procedure, as it should be quite hard to find the right parameters to make it converge).

Reviewer 3



This paper presents the limitations of policy gradient-based methods which need the explicit p.d.f of action in continuous control, and gives the proof that the Gaussian distribution strategy cannot converge to the optimal under some conditions. Then it introduces the DPO framework that can converge to an optimal solution without the requirement of the underlying p.d.f and thus without the limitation of parametric distribution. Also, the paper presents a practical algorithm GAC that applies Quantile Regression and Autoregressive Implicit Quantile Networks which can represent arbitrarily distributions. GAC achieves good results in continuous control tasks and some are better compared to the policy gradient baselines, and it has the same efficiency but requires more computation. Minor issues: The description of ‘the sub-optimality of uni-modal policies’ (from line 73) is confusing. Which part of figure 1a corresponds to ‘the predefined set of policies’? In line 76, ‘this set is convex in the parameter µ’ seems to mean ‘this set is convex in the parameter space Θ’, and what does ‘it is not convex in the set Π’ mean? What is the definition of ‘(1−α)δ µ 1 +αδ µ 2’? The condition at the end of line 96 seems not written properly. Should the right side of the equation in line 99 (also 442) be 1-2ε and ε<1/3?

[Author Response · NeurIPS 2019]

We thank the reviewers for their interest in our work and their helpful comments. Please find our response below.

**Comments relevant to all reviewers:**

- In the three time scale procedure, faster time scales view slower time scales as static. This is why the fastest time scale
is essentially solving a supervised learning problem over two static networks.

- The slowest time scale (delayed actor) is required both theoretically as well as empirically. Without it, convergence is
not guaranteed and the algorithm becomes unstable.

**Reviewer 1:**

- You are correct in pointing out that an on-policy version of Algorithm 1 is not ensured to converge. DPO is an
off-policy actor-critic framework which requires that all state action pairs are visited "enough" in order to ensure
convergence, which is a theoretical assumption in various off-policy algorithms. We achieve this, similar to DQN,
DDPG and TD3, by keeping an exploration strategy which does not decay to zero. We will emphasize this to avoid
confusion in the paper.

- Your observation is correct, DPO and GAC are not perfectly aligned. DPO requires optimization in the distribution
space, while GAC is a practical approximation of DPO in which optimization occurs in the space of parameters of the
generative model. The DPO framework is a fundamental framework which can be extended in a similar way as Policy
Gradient methods to bridge the gap between DPO and GAC.

- GANs and VAEs are definitely a valid choice for representing the policy, yet they have some pitfalls [1]. GANs pose
the problem of learning a generative model and solving a two-player zero-sum game. This form of learning in itself
is often unstable (resulting in mode-collapse) and still lacks theoretical guarantees and stability assurances. VAEs
minimize the KL distance, as opposed to the p-Wasserstein distance, which has its own benefits. The quantile approach
overcomes these issues by directly minimizing the p-Wasserstein distance using the quantile regression loss.

**Reviewer 2:**

- This is a good point. In practice, the three time-scale requirement is implemented using different learning rates so that
the various elements converge at different rates. We follow a similar implementation method as in other actor-critic
approaches, which are based on two timescales.

- *Derivation of policy distribution update:*
What we ultimately wish to have is an update similar to that of Policy Iteration (or more specifically, Approximate
Policy Iteration), which conservatively updates the policy given a target policy $\pi'$ as:
$$\pi_{k+1}(a|s) = (1 - \alpha_k)\pi_k(a|s) + \alpha_k\pi'(a|s).$$
Policy Iteration schemes use the target policy $\pi'(a|s) \in \arg\max_{a \in \mathcal{A}} r(s,a) + \gamma \sum_{s' \in \mathcal{S}} P(s'|s,a)v^{\pi_k}(s')$ in the exact
case, or the $\epsilon$-greedy target policy in the approximate case. Nevertheless, finding the $\arg\max$ is itself a hard problem in
non-convex continuous regimes. Since finding the greedy action is complicated, it is reasonable to instead define the
target policy (i.e., $\pi'$) as a distribution over all improving actions. We denote this target policy by $\pi'(a|s) = \mathcal{D}_{I^\pi}^{\pi}(a|s)$.
Finally, we take a gradient based approach using a distance metric over policies, $d$, yielding the DPO update rule
$$\pi_{k+1} = \Gamma\left(\pi_k - \alpha_k \nabla_\pi d(\mathcal{D}_{I^{\pi_k}}^{\pi_k}, \pi) \mid_{\pi=\pi_k}\right).$$

**Reviewer 3:** Thank you for pointing out some confusing explanations, we will make sure to clarify them in the paper.

- In Fig. 1a the intention is to compare optimization w.r.t. the policy itself (i.e., $\nabla_\pi v^\pi$) to optimization w.r.t. the
parametrization $\nabla_\theta v^{\pi_\theta}$ (e.g., for Delta distributions $\theta$ represents the action, and for Gaussian distributions $\theta$ represents
the mean $\mu$). The former is what classical algorithms such as CPI (Kakade and Langford 2002) require, whereas the
latter is what occurs in the standard policy gradient approaches. In Fig. 1a, the left ($\Pi$ space) represents the ideal
approach, whereas the right ($\Theta$ space) represents the sub-optimality which occurs when encountering a non-convex
action-value function and optimizing with respect to the parametric distribution parameters (e.g., action).

- Regarding convexity, our intent was to show that the set $\Theta$ is not convex in a probabilistic sense. The set $\Pi$ is the set
of all probability distributions, whereas $\Theta$ is the span of probabilities distributions that $\pi_\theta$ can represent. Gaussian or
Delta distributions are limited to their set, and thus can't ensure convergence to a global extrema. More specifically,
$\alpha\delta_{\mu_1} + (1-\alpha)\delta_{\mu_2}$ means to play action $a_1 = \mu_1$ with probability $\alpha$ and $a_2 = \mu_2$ otherwise, but this distribution is not
a Delta distribution, and therefore is not contained in $\Theta$ - the set of all Delta distribution functions. We will make this
notation clear in the paper.

- Thank you for noting the mistake in lines 99 and 442 - we've updated the paper.

[1] Georg Ostrovski, Will Dabney, and Remi Munos. Autoregressive quantile networks for generativemodeling. *arXiv*
*preprint arXiv:1806.05575*, 2018.


[Meta-Review · NeurIPS 2019]

The reviewers were in agreement that this is an interesting paper, though lacked in clarity in places. The empirical results are reasonably strong.